# Integrating Whole-Genome Resequencing and RNA Sequencing Data Reveals Selective Sweeps and Differentially Expressed Genes Related to Nervous System Changes in Luxi Gamecocks

**DOI:** 10.3390/genes14030584

**Published:** 2023-02-25

**Authors:** Jieke Zhou, Ying Chang, Junying Li, Haigang Bao, Changxin Wu

**Affiliations:** National Engineering Laboratory for Animal Breeding, College of Animal Science and Technology, China Agricultural University, Beijing 100193, China

**Keywords:** whole-genome resequencing, selection signature, RNA sequencing, cerebral cortex, midbrain, nervous system

## Abstract

The Luxi gamecock developed very unique morphological and behavioral features under the special artificial selection of the most famous Chinese gamecocks. There are very few research studies on the genetics and selection of the Luxi gamecock. We used six methods (Fst, Tajima’s D, hapFLK, iHS, XP-EHH, and Runs of homozygosity) to detect selective sweeps in whole-genome resequencing data of 19 Luxi gamecocks compared to other Chinese indigenous chickens. Eleven genes that were highly related to nervous system development (*CDH18*, *SLITRK1*, *SLITRK6*, *NDST3*, *ATP23*, *LRIG3*, *IL1RAPL1*, *GADL1*, *C5orf22*, *UGT8*, *WISP1*, and *WNT9A*) appeared in at least four methods and were regarded as the most significant genes under selection. Differentially expressed gene (DEG) analysis based on the RNA sequencing data of the cerebral cortex and middle brain between six Luxi gamecocks, Tibetan chickens, and white leghorns found that most differentially expressed genes were enriched in pathways with nervous system functions. Genes associated with aggressiveness-related neurotransmitters (*SLC4A2, DRD1, DRD2, ADRA2A*, and *ADRA2B*) showed differential expression rates in Luxi gamecocks as well. Combined results showed that most genes in selective sweep regions were also differentially expressed in Luxi gamecocks including the most significant genes (*SLITRK6, IL1RAPL1, GADL1, WISP1*, and *LRIG3*). This study provides more insight into molecular mechanisms of the aggressiveness of gamecocks and aims to promote further studies on animal and human aggression.

## 1. Introduction

Gamecocks have been selected for entertainment for at least 2000 years [1]. It is not only an ornamental chicken breed but is also capable of breeding new commercial chickens with high performance. China has several native gamecock breeds, including the Luxi gamecock, the Henan gamecock, the Tulufan gamecock, the Xishuangbanna gamecock, and the Zhangxi gamecock, among others [2]. These breeding resources mostly remain unexploited until now, and we know little about the genetics of the unique and economic traits of cock fighting. Mitochondrial DNA revealed genetic relationships among Chinese gamecocks [3,4] and suggested that gamecocks may be the earliest domesticated chicken in China. Guo et al. [5] reported some selected genes whose functions were associated with organ development, aggressive behaviors, and energy metabolism in the Xishuangbanna gamecocks compared to the Tibetan chicken and red jungle fowl via several selection signature methods. Until 2020, Luo et al. [6] used whole-genome resequencing data to reveal the genome diversity of Chinese gamecocks and the genetic relationships between gamecocks and other China native chickens and commercial lines. Several selective sweeps were found that may be related to the aggressiveness of Chinese gamecocks, such as alkylglycerol monooxygenase (*AGMO*), isoprenoid synthase domain-containing protein (*ISPD*), and carboxypeptidase Z (*CPZ*). Komiyama et al. [7,8,9,10,11,12] completed a series of research studies on the Japanese gamecock intending to illuminate its origin, as well as its relationships with Japanese indigenous chickens and candidate genes concerning aggressiveness. The research found that the Japanese gamecock may be the origin of other Japanese indigenous chickens [12] and that selective sweeps were related to aggressiveness, including the dopamine receptor (*DRD*) gene family in the Japanese gamecock [7,8]. Li et al. [13] detected 33 significant SNPs related to aggressiveness in chickens in a genome-wide association study, the most significant of which was the one located in the intron region of the sortilin-related VPS10 domain-containing receptor 2 (*SORCS2*) gene. However, all these studies only found selection sweeps in genomes, and the expression situations of these genes were under-represented.

Studies on animal aggressiveness already found that three main neurotransmitters affected aggressiveness: serotonin, noradrenaline, and dopamine [13,14,15,16,17,18,19,20,21]. Usually, serotonin inhibits an animal’s aggressiveness [14,15]. Animals with high levels of serotonin are tamer than those with low levels [14,15,16,17,18]. For example, the silver fox selected for low levels of defensive aggressive behavior showed higher serotonin levels than the wild breed [14]. In mice experiments, high levels of serotonin were also associated with less aggression [14,15,16,17]. In humans, low levels of serotonin were also highly associated with aggressive and violent behaviors [18]. In contrast to serotonin, noradrenaline levels showed a positive relation with aggressiveness in both human and nonhuman animals [19]. High levels of noradrenaline were also seen to make people more aggressive when faced with external stimulators [19]. On the other hand, mice were found to have less aggressive behaviors with low levels of noradrenaline [20]. Komiyama et al. [10] tested the concentrations of four monoamine neurotransmitters and six derived metabolites in the cerebral cortex, midbrain, and striatum of the Japanese gamecock and shaver brown chickens, and found that only the noradrenaline level in the midbrain of the Japanese gamecock was significantly higher. In most cases, a high level of dopamine will increase an animal’s aggressiveness [14], just like noradrenaline. In hens [21] and green anoles [22], studies found that the excitation of dopamine receptor type-1 and type-2 (*DRD1* and *DRD2*) will impair aggressive behaviors. Li et al. [13] found that *SOCRS2* genes were highly associated with chicken aggressiveness and its knock-down decreased the expression level of *DRD1*, *DRD2*, *DRD3*, and *DRD4* genes.

The Luxi gamecock is representative of Chinese gamecocks and is mainly spread over the Shandong province of China. In this study, our main goal was to find whether entertainment fighting selection changes nervous system functions in Luxi gamecocks, especially genes related to serotonin, noradrenaline, and dopamine. We wish to narrow down the scope of candidate genes by integrating whole-genome resequencing data and RNA sequencing data. We sampled 19 Luxi gamecocks for whole-genome resequencing and collected another 120 WGS data sets from five Chinese indigenous chicken breeds [5,23,24] to detect selective sweeps in the Luxi gamecock. To minimize false positives and to obtain a comprehensive set of selective sweeps, we combined five methods that use all three pieces of genome information to obtain more accurate and reliable results (Tajima’D is based on an allele frequency spectrum, iHS and XP-EHH are based on linkage disequilibrium haplotypes, and Fst and hapFLK are based on population differentiation). Then, we extracted total RNA from the cerebral cortex and midbrain samples of six Luxi gamecocks, six Tibetan chickens, and six white leghorns to perform differential gene expression analysis and screen potential genes which may influence nervous system development and aggressiveness-related neurotransmitters. Finally, we observed overlapping genes between selective sweeps and differentially expressed genes and found that selection for entertainment fighting indeed changed the nervous system of the Luxi gamecock, including genes relating to three neurotransmitters.

## 2. Materials and Methods

### 2.1. DNA Sampling and Genome Resequencing Data

The Luxi gamecock came from the Poultry Genetic Resources and Breeding Station of China Agricultural University. In total, 19 blood samples were collected from the wing veins of 19 Luxi gamecocks (about 30 weeks old) using 1 mL injectors. The total DNA of each blood sample was extracted using a genomic DNA Kit (Cat. #DP304-03, TIANGEN Biotech (Beijing) Co., Ltd., China) according to the manufacturer’s instructions. After the DNA samples were checked and quantified, they were delivered to a commercial company (Novogene Co., Ltd., Beijing, China) for whole-genome resequencing. The datasets with paired-end reads were produced on an Illumina NovaSeq 6000 platform. The average depth of resequencing for each sample was 10×. The other 106 samples’ whole-genome sequencing data of 5 chicken breeds were downloaded from the NCBI and the ChickenSD database, as shown in Table 1. 

Raw data quality was cleaned by Trimmomatic-0.39 [25] with the following commands: LEADING:3, TRAILING:3, HEADCROP:10, SLIDINGWINDOW:4:15, and MINLEN:70. The statistics of the clean data quality were assessed by FastQC (version 0.11.9) [26]. 

### 2.2. Mapping and SNP Calling

Cleaned data were mapped to the chicken reference genome GRCg6a using BWA (version 0.7.17) [27] with the mem algorithm. The picard component in GATK (version 4.1.8.1) [28] was employed to flag duplicated reads. SAM files generated by BWA were reordered by chromosome and converted to the BAM format with samtools (version 1.11) [29].

SNP calling was performed in line with a genome analysis toolkit (GATK) pipeline [28]. A set of known SNP sites of chicken was downloaded from Ensemble (http://ftp.ensembl.org/pub/release-104/variation/gvf/gallus_gallus/gallus_gallus.gvf.gz, accessed on 5 August 2021) to perform the BaseRecalibrator process in GATK and to generate the recalibrated BAM files for each individual. The joint-calling strategy of GATK was chosen to obtain more accurate results and find rare variants. Firstly, each individual generated a gvcf format mediate file via the HaplotypeCaller command of GATK. Secondly, the GenomicsDBImport command was used to create a temporary database that combined all 258 individuals’ GVCF results, divided by chromosomes. Finally, we used the GenotypeGVCFs command to obtain the final SNP calling results in a VCF file, including all samples.

After SNP calling, the results were filtered by hard criteria recommended by GATK: QD > 2.0, FS > 60.0, MQ > 40.0, MQRankSum > −12.5, ReadPosRankSum > −8.0, SOR > 3.0. The filtered SNP data were then phased and imputed by BEAGLE (version 5.2) [30] with default parameters, and only the autosome sites were retained for subsequent analysis. All identified SNPs were annotated by snpEff [31] software (latest core version accessed on 25 August 2021).

Before downstream analysis, we converted the final VCF file to a binary plink format via PLINK (version 1.9) [32]. Variants were pruned according to linkage disequilibrium (LD) to reduce potential basis by using the “–indep-pairwise 0.5” command. 

### 2.3. Detection of Selective Sweeps

The selective sweep detection method usually uses three kinds of genome information: an allele frequency spectrum, a linkage disequilibrium haplotype, and population differentiation. To obtain the most powerful efficiency rates and reduce false-positive rates as much as possible, we chose five methods with all three kinds of information, including fixed index (Fst) [33] and hapFLK (based on population differentiation [34]), Tajima’s D (based on the allele frequency spectrum) [35], as well as the integrated haplotype score (iHS) [36] and the cross-population extended haplotype score (XP-EHH) (based on the linkage disequilibrium haplotype [37]). We also used runs of homozygosity (ROHs) [38] to detect selective sweeps, which is a more recent method for finding selective sweeps. Fst and Tajima’s D values were calculated by VCFTools (version 0.1.19) [39]. Fst was calculated in a sliding window of 40 kb size with a step of 20 kb. Tajima’s D values were calculated in a non-overlapping 10 kb sliding window. hapFLK [34] was completed with a python script provided by the author with cluster parameter *k* = 12. Calculations of iHS and XP-EHH were run by selscan [40] with default parameters. Only negative Tajima D, iHS, and XP-EHH values were retained for subsequent analysis. For methods except ROH, the top 1% of outliers were regarded as significant sites that may be putative sites or regions under selection. For ROH, SNPs that occurred in a ROH with high frequencies, specifically in LXFCs, were regarded as selective sites. Genes identified by at least two methods were considered as most likely putative genes under selection. 

### 2.4. RNA Extraction and Sequencing

The heads of six chickens from each of the Luxi gamecock, Tibetan chicken, and White Leghorn were rapidly cut off to enable sampling of the cerebral cortex and midbrain in chickens that were 64–66 weeks old. The total RNA was isolated by trizol methods and then sent to Novogene Co. Ltd to sequence on the Illumina NovaSeq 6000 platform. Raw RNA data were filtered by Fastp (version 0.20.1) [41] with the following commands: -f 19, -F 19, -q 20, -c -w 6, -l 50, -n 6, and -z 6. They were then evaluated by fastQC (version 0.11.9) [26]. Hisat2 (version 2.2.1) [42] was used to build an index and map clean reads to the chicken reference genome GRCg6a, and the results were converted to BAM files by samtools (version 1.11) [29]. Data accessibility could be found in Appendix A.

### 2.5. Gene Quantification and Differential Gene Expression Analysis

Gene expression abundance was estimated by featureCOUNTS [43] with FPKM (fragment per kilobase of transcript per million mapped reads). Differentially expressed genes (DEGs) between LXFCs and TCs or the White Leghorn (WL) chickens were detected by the DESeq2 [44] package of R. Differentially expressed genes were then detected by the threshold of an adjusted p-value (padj) below 0.05. Values of log2fold change larger than 1 or less than −1 were regarded as differentially expressed genes. Shared DEGs between LXFCs and TCs and between LXFCs and WL chickens were chosen for GO and KEGG pathway enrichment analysis. We further observed the genes that shared differential expression and were within a selective sweep region.

### 2.6. Gene Ontology Enrichment and Pathway Analysis

All annotated genes were submitted to g:Profiler [45] (https://biit.cs.ut.ee/gprofiler/gost, archived version Ensembl 106, Ensembl Genomes 53, built on 18 May 2022, accessed on 8 June 2022) for function and pathway enrichment analysis with the *Gallus gallus* organism background and a 0.05 threshold.

## 3. Results

### 3.1. Overview of the Whole-genome Sequencing Data and the RNA-Sequencing Data

After variant processing, we identified 43,259,995 SNPs and 5,488,914 indels that passed the hard-quality GATK filter of the GATK. In this study, we only kept biallelic SNPs for subsequent analysis, which excluded 2,101,476 SNP sites and kept 41,158,519. The SNPs that remained were then phased and imputed by BEAGLE with default parameters. Next, we pruned data with a linkage disequilibrium threshold of *r*^2^ = 0.5. As a result, only 24,697,629 SNPs remained.

After the quality control and mapping of raw RNA sequencing data, alignment rates ranged from 85.66% to 92.35%, with an average rate of 90.82%. Detailed alignment ratios are shown in Appendix A.

### 3.2. Selective Sweeps Revealed Genes Are Mostly Associated with Organism Development

We used five methods (Fst [33], Tajima’D [35], hapFLK [34], his [36], and XP-EHH [37]) to detect selective sweeps in LXFCs compared to other domesticated chickens. We kept selective sweeps, at least those detected in two methods, to improve accuracy and reduce false-positive results. The top 1% of outliers were detected as selective sweeps and are shown in Figure 1. Finally, 2197 genes located in putative selective regions were found. Genes without a clear functional annotation were excluded and 363 genes remained, the list could be found in Appendix A. These 363 genes were mostly associated with organism development, according to GO enrichment results, including those associated with the nervous system, which were expected to undergo selection and organ, skeleton, and embryo development, as shown in Figure 2. Eleven genes listed in Table 2 were detected by at least four methods and regarded as the most significant selective sweeps. The only gene shared by all five methods is *CDH18* (*cadherin 18*), whose function was found to be highly related to organism development [46,47]. Moreover, at least four of the eleven genes were found to be related to nervous system function: *SLITRK1* [20,48,49], *SLITRK6* [50,51], *IL1RAPL1* [52], and *NDST3* [53,54]. Another three were reported to participate in nervous system development and function: *WISP1* [55], *UGT8* [56,57], and *LRIG3* [58,59]. Besides these eleven genes, we also observed *ADRA1D* and *ADRA2A,* whose functions are related to noradrenaline pathways, and many SLC gene family members that encode neurotransmitter transporters [60] in the cerebrum.

Furthermore, the SNPs identified by the ROH method also found two unique outstanding peaks in the Luxi gamecock: the 5.7–6.3 Mb region of Chr3 and the 0.43–1.24 Mb region of Chr8 (Figure 1F). No functional genes were annotated in the former region, while only two functional genes were found in the latter: *AMY2A* (amylase, alpha 2A) and *NTNG1* [61,62] (netrin G1), the latter of which is another gene associated with the nervous system.

### 3.3. Differentially Expressed Genes between LXFCs and TC/WL Chickens Were Significantly Associated with Nervous System in the Midbrain

Differentially expressed genes were detected according to FPKM via the DESeq2 package. Between LXFC and TC birds, 735 down-regulated genes and 442 up-regulated genes were detected in the cerebral cortex of LXFCs compared to TC birds, while 3875 down-regulated genes and 3759 up-regulated genes were detected in the midbrain of LXFCs compared to TCs. When comparing LXFC and WL chickens, 1791 down-regulated genes and 1861 up-regulated genes were detected in the cerebral cortex, while 4606 down-regulated genes and 4771 up-regulated genes were detected in the midbrain. Heat maps of the four data sets are shown in Figure 3. The shared differentially expressed genes shown in Figure 4 contained 417 down-regulated genes and 220 up-regulated genes in the cerebral cortex of LXFCs, as well as 3173 down-regulated genes and 3296 up-regulated genes in the midbrain of LXFCs. 

DEGs in the cerebral cortex showed no significant GO enrichment terms. However, DEGs in the midbrain were enriched in several pathways and are shown in Figure 5. Down-regulated DEGs were mainly associated with ribosome, oxidative phosphorylation, and proteasome (Figure 5A). Up-regulated DEGs were associated with various signaling pathways (Figure 5B) related to nervous system function and development, such as the ErbB signaling pathway [63,64], the MAPK signaling pathway [65], and the mTOR signaling pathway [66,67,68].

We notably observed *solute carrier family 6 member 4* (*SLC6A4*) which encodes the transporters of serotonin; two types of dopamine receptor genes (*dopamine receptor D1* (*DRD1*) and *dopamine receptor D2* (*DRD2*)) and one noradrenaline receptor gene *adrenoceptor alpha 2A* (*ADRA2A*) were up-regulated in the midbrain of LXFCs compared with TC and WL chickens. The log2fold-change values of these genes were 2.14, 0.57, 0.70, and 0.68, respectively, compared to TC chickens, and 2.17, 1.15, 1.30, and 0.64, respectively, when compared to WL chickens. Another noradrenaline receptor gene, adrenoceptor alpha 2B *(ADRA2B*), a shared down-regulated gene in the cerebral cortex, showed a −1.57 log2fold-change compared to TC and −0.7 compared to WL chickens. *ENSGALG00000037163*, a sodium-dependent serotonin transporter-like gene, was a down-regulated gene in the midbrain of LXFCs which showed a log2fold-change of −2.99 (compared to TC) and −1.52 (compared to WL chickens). All DEGs used for each comparison could be found in Appendix A. 

### 3.4. Genes under Selective Sweeps Mostly Showed Differential Expression between LXFC and TC/WL Chickens in the Midbrain

We further observed results shared between selective sweeps and DEGs. In the cerebral cortex, 19 genes were found to be shared between the two analyses of LXFC and TC chickens, with 84 genes shared between analyses of LXFC and WL chickens. In the midbrain, 187 shared genes were found between LXFC and TC birds, with 214 genes shared between LXFC and WL chickens. Seven genes of the eleven genes in Table 2 were shared genes, including *SLITRK6*, *IMMP2L*, *NDST3*, *LRIG3*, *WISP1*, *NTNG1*, and *IL1RADL1*. These results are shown in Figure 6

Shared DEGs in the cerebral cortex did not gain significant results in GO and KEGG enrichment. Shared DEGs in the midbrain were highly correlated with organism development, including the nervous system (Figure 6). Detailed gene list could be found in Appendix A.

## 4. Discussion

In the present study, 19 Luxi gamecocks and 120 other samples (representing 5 breeds) were used to detect selective sweeps in the Luxi gamecock. Six Luxi gamecocks, Tibetan chickens, and white leghorns were used to conduct RNA sequencing. To gain the highest detecting power, we decided to use six methods to detect selective sweep: Tajima’s D [35], iHS [36], XP-EHH [37], Fst [33], and hapFLK [34]. Moreover, ROH information could also be used in selective sweep detection [38]. To reduce false-positive results, we only kept genes detected by at least two methods as significant genes. Eventually, 363 genes remained. GO and KEGG pathway enrichment results only showed BP terms (see Figure 2). Most BP terms were mainly associated with organism development and morphogenesis, such as the nervous system, embryo, skeleton, and organs. Among these 363 genes, 11 genes were seen as the most significant since they were detected by at least four methods, and most genes were very likely to participate in nervous system development, such as *SLITRK1* [20,48,49], *SLITRK6* [50,51], *IL1RAPL1* [52], *WISP1* [55], *UGT8* [56,57] and *LRIG3* [58,59]. Meanwhile, another two genes, *AMY2A* and *NTNG1*, were determined by ROH. It has also been reported that *NTNG1* is associated with nervous system disease [61,62]. *SLITRK1* had been found to be related to a genetic nervous disorder called Tourette’s syndrome [48,49]. *SLITRK1* knock-out mice showed increased anxiety-like behaviors and abnormal noradrenergic activity, and also had higher levels of norepinephrine and metabolite 3-methoxy-4-hydroxyphenylglycol [20]. *SLITRK6* regulates the outgrowth of sensory neurons in the inner ear and retina; its mutation may cause myopia and deafness [50,51]. *NDST3* [53,54] and *NTNG1* were both found to be highly associated with schizophrenia in humans [61,62,69]. *IL1RAPL1* can affect the morphogenesis of cortical neurons in the spine and significantly change the behaviors of mice [52]. *IL1RAPL1* knock-out mice had shorter fear memory, more active social behaviors, and decreased cortical neurons density in the spine [52]. *WISP1* could prevent neuron injuries and could become a target in therapeutic strategies for some disorders such as traumatic injury [55]. Though the detailed function of *UGT8* is unclear, it is highly expressed in the brain [56,57]. Some researchers reported connections between *LRIG3* and the nervous system [59]. Moreover, among 363 significant selective genes, we overserved adrenoceptor alpha 1D (*ADRA1D*), adrenoceptor alpha 1A (*ADRA1A*), inner mitochondrial membrane peptidase subunit 2 (*IMMP2L*), carboxypeptidase Z (*CPZ*), and several numbers of the SLC gene family (solute carrier family). *ADRA1D* and *ADRA1A* encoded two kinds of adrenaline receptors. Meanwhile, noradrenaline has long been known to be related to aggressiveness in animals [70,71]. *IMMP2L* was another target gene of Tourette’s syndrome [72] and autism [73]. *CPZ* had already been reported to be related to aggressive behaviors in chickens [6,13]. Many SLC family (solute carrier family) genes (*SLC35F5, SLC38A2, SLC39A6, SLC4A11, SLC4A1AP, SLC30A8, SLCO1A2*, and *SLC9A7*) are mainly expressed in the brain to produce transporters of various neurotransmitters [60]. Various results suggest that the nervous system of the Luxi gamecock was significantly affected by entertainment fighting selection, including genes directly associated with three aggressiveness-related neurotransmitters (serotonin, noradrenaline, and dopamine).

Considering the tight relationships between selective sweeps and the nervous system, we decided to sample the cerebral cortexes and midbrains of six Luxi gamecocks, six Tibetan chickens, and six white leghorns to obtain RNA sequencing datasets. We tried to explore whether expression patterns in the nervous system of the Luxi gamecock changed compared with other chickens. Other results show that most DEGs appeared in the midbrain of LXFC chickens, rather than the cerebral cortex, where serotonin [14,15,74] and dopamine [15] are produced. GO and KEGG enrichment results of down-regulated genes in the midbrain of LXFC chickens did not show obvious connections with the nervous system. On the other hand, up-regulated genes were enriched in terms involved in nervous system development (see Figure 5). Enriched pathways were mostly associated with the nervous system, including ErbB signaling pathways [63,64], mTOR signaling pathways [66,67,68], and MAPK signaling pathways [65]. We specifically found a group of genes associated with dopamine, serotonin, and noradrenaline which showed significant differential expressions in the midbrain of LXFC birds such as *SLC6A4*, *ENSGALG00000037163*, *DRD1*, *DRD2*, *ADRA2A*, and *ADRA2B*. *SLC6A4*, up-regulated in the midbrain of LXFC birds, encodes serotonin transporters in the central nervous system [60], while another serotonin transporter-like gene, *ENSGALG00000037163*, was found to be down-regulated in the midbrain of LXFCs. *DRD1* and *DRD2* encode two types of dopamine receptors. Multiple animal research proved that the excitation of dopamine receptors 1 and 2 would impair aggressive behaviors [21,22,75]. There are two types of adrenaline receptors (*ADRA2A* was up-regulated while *ADRA2B* was down-regulated in the midbrain of LXFCs). Usually, there is a positive correlation between the noradrenaline level and aggressiveness [19], but a negative correlation between the adrenaline receptor alpha 2 subtype has also been reported [19].

We found that selective sweeps and differential gene expression analysis were both related to the nervous system in the brain of LXFCs. To explore further connections between selective sweep regions and DEGs, we observed the shared genes between the two analyses. Few selective sweeps showed a DEG in the cerebral cortex, while most selective sweeps were also DEGs in the midbrain of LXFCs. About 51% (187/363) of genes under selective sweeps were also DEGs in the midbrain of LXFC compared to TC chickens, while 59% (214/363) of genes under selective sweep regions were also DEGs in the midbrain of LXFC birds when compared to WL chickens. Six of the eleven most significant genes under selective sweeps included the following: *SLITRK6*, *NDST3*, *IL1RAPL1*, *WISP1*, *LRIG3*, and *NTNG1*. Most of these also influence the development and function of the nervous system. The results provide us new insight for further research on the genetics of aggressiveness in Luxi gamecocks. Future work will focus on how these genes are expressed in different breeds and at different times, as well as their relationship with the three aggressiveness-related neurotransmitters.

## 5. Conclusions

In this study, we first sequenced the cerebral transcriptome of the Luxi gamecock and combined it with whole-genome resequencing data. The results confirmed that the nervous system of the Luxi gamecocks changed under selection, and we located several candidate genes that indirectly or directly influence three neurotransmitters. We combined six methods to detect selective sweeps in Luxi gamecocks and found 11 significant genes that were mostly related to nervous system development. Furthermore, 417 down-regulated genes and 220 up-regulated genes were found in the cerebral cortex of Luxi gamecocks, while 3173 down-regulated genes and 3296 up-regulated genes were found in the midbrain of Luxi gamecocks. Most were enriched in pathways that were associated with nervous system functions. More than half of the genes in selective sweep regions also showed differential expression. Combined results suggest that the artificial selection for entertainment fighting significantly changed the development and function of the nervous system of Luxi gamecocks. Genes such as *IL1RAPL1*, *SLITRK6*, *SLC6A4*, *DRD1*, *DRD2*, *ADRA1D*, and *ADRA2A* were important candidate genes that affected the aggressiveness of Luxi gamecocks. This study provided more insight into the genetics of the aggressiveness of gamecocks and will inform further research on the genes that affect the aggressiveness of animals and humans.

## Figures and Tables

**Figure 1 genes-14-00584-f001:**
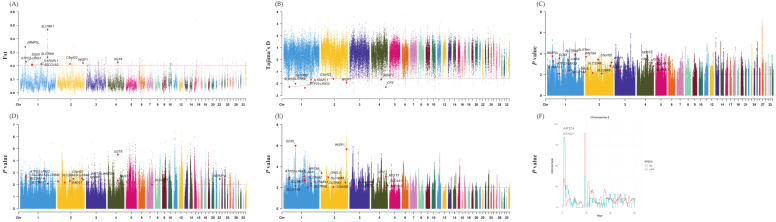
Manhattan plots generated by six selective sweep detection methods. (**A**) Fst values. (**B**) Tajima’s D values. (**C**) experienced distribution p values of hapFLK. (**D**) experienced distribution *p* values of iHS. (**E**) experienced distribution *p* values of XP-EHH. (**F**) SNP occurrence percentage on Chr8. DCs refer to domesticated chickens except LXFCs.

**Figure 2 genes-14-00584-f002:**
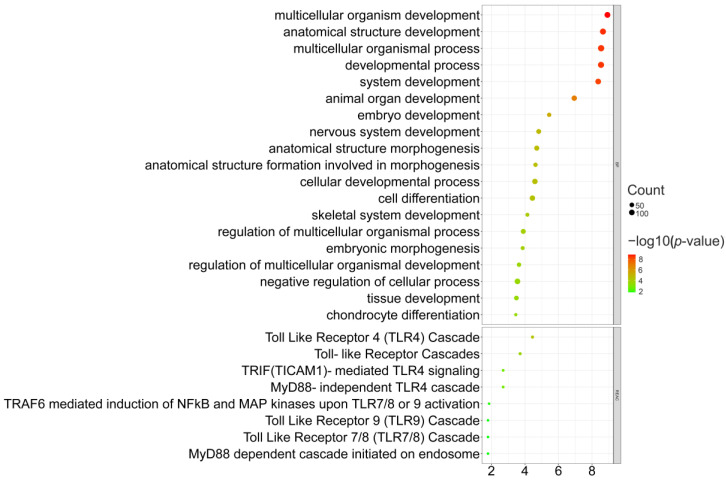
GO and KEGG pathway enrichment results of selected genes detected by at least two methods.

**Figure 3 genes-14-00584-f003:**
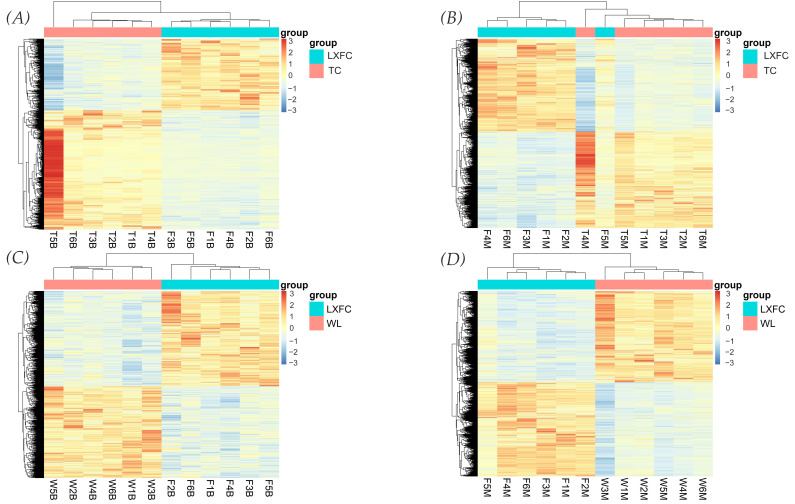
Heatmaps of genes differentially expressed between LXFC and TC/WL chickens in the cerebral cortex and midbrain. (**A**) DEGs between the cerebral cortex of LXFC and TC birds. (**B**) DEGs between the midbrains of LXFC and TC birds. (**C**) DEGs between the cerebral cortex of LXFC and WL chickens. (**D**) DEGs between the midbrains of LXFC and WL birds.

**Figure 4 genes-14-00584-f004:**
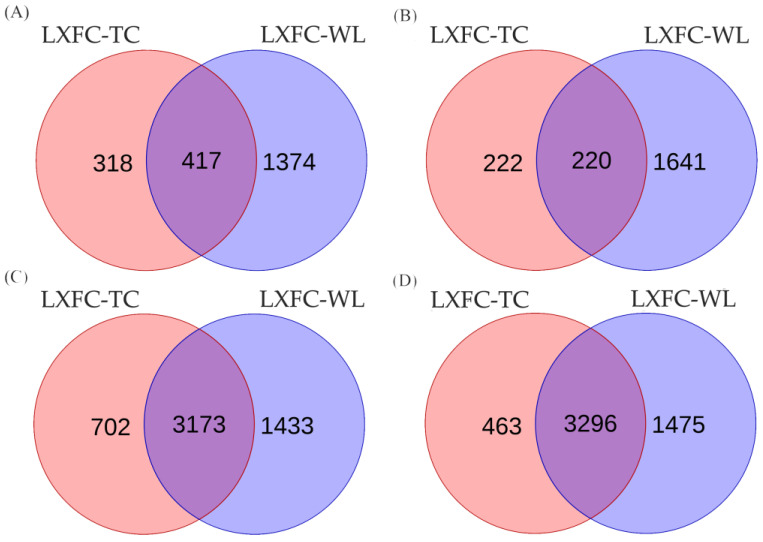
Venn diagrams of shared DEGs between the cerebral cortex and midbrain of LXFC-TC chickens and LXFC-WL chickens. (**A**) Shared down-regulated DEGs in the cerebral cortex between LXFC-TC and LXFC-WL chickens. (**B**) Shared up-regulated DEGs in the cerebral cortex between LXFC-TC and LXFC-WL chickens. (**C**) Shared down-regulated DEGs in the midbrain between LXFC-TC and LXFC-WL chickens. (**D**) Shared up-regulated DEGs in the midbrain between LXFC-TC and LXFC-WL chickens.

**Figure 5 genes-14-00584-f005:**
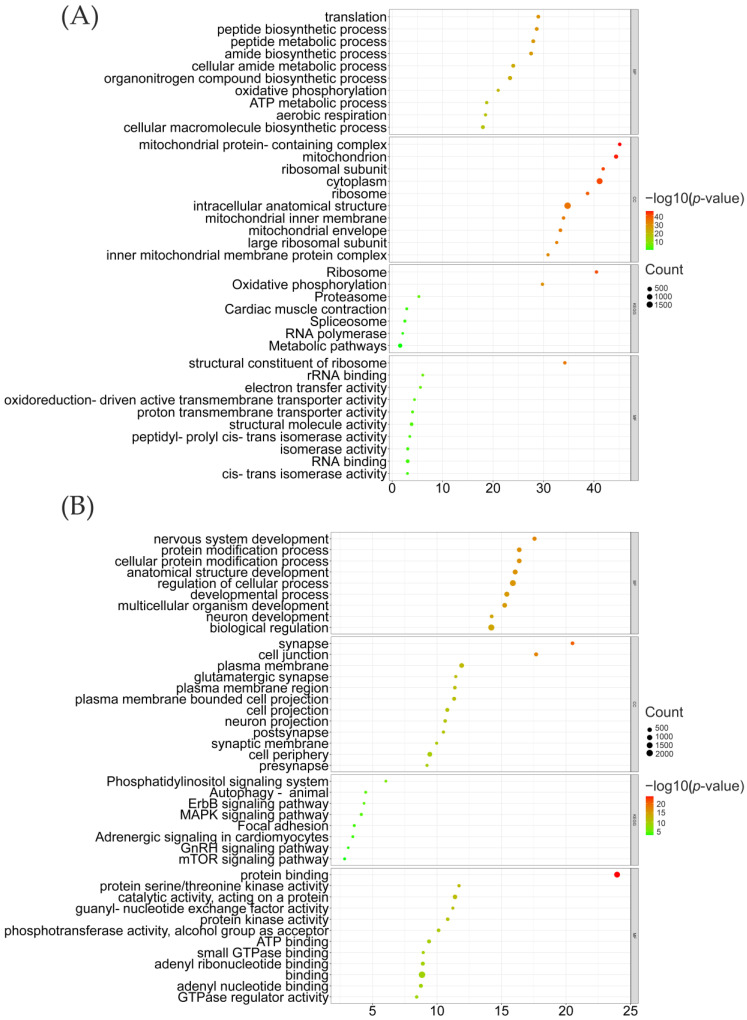
GO and KEGG enrichment results of midbrain DEGs shared between LXFC-TC and LXFC-WL chickens. (**A**) shared down-regulated DEGs (**B**) shared up-regulated DEGs.

**Figure 6 genes-14-00584-f006:**
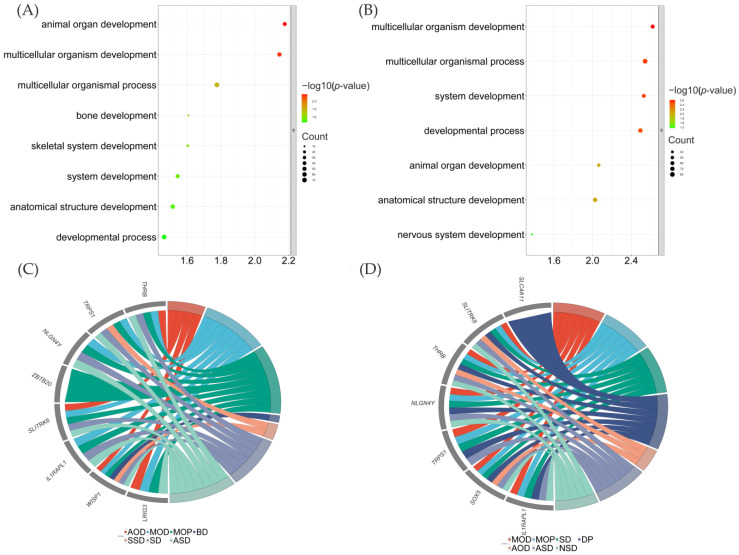
GO enrichment of genes shared between selective sweeps and DEGs in the midbrain. (**A**) LXFC-TC chickens. (**B**) LXFC-WL chickens. (**C**) Chord diagram of overlapping genes between LXFC-TC birds. (**D**) Chord diagram of overlapping genes between LXFC-WL chickens. AOD refers to animal organ development; MOD refers to multicellular organism development; MOP refers to multicellular organismal process; BD refers to bone development; SSD refers to skeletal system development; SD refers to system development; ASD refers to anatomical structure development; DP refers to developmental process; AOD refers to animal organ development; NSD refers to nervous system development.

**Table 1 genes-14-00584-t001:** Populations used for whole genome resequencing.

Population	Abbreviation	Numbers
Luxi gamecock	LXFC	19
Shandong native chicken	SDNC	30
Xishuangbanna gamecock	YNFC	16
Yunnan native chicken	YNNC	22
Tibetan chicken	TC	18
Rhode Island red	RDH	20

**Table 2 genes-14-00584-t002:** Selective sweeps detected by four methods.

Gene	Full Name	Fst	Tajima’s D	hapFLK	iHS	XP-EHH
*CDH18*	cadherin 18	√	√	√	√	√
*SLITRK1*	SLIT and NTRK-like family member 1	√	-	√	√	√
*SLITRK6*	SLIT and NTRK-like family member 6	√	√	√	√	-
*NDST3*	N-deacetylase and N-sulfotransferase 3	-	√	√	√	√
*ATP23*	ATP23 metallopeptidase and ATP synthase assembly factor homolog	√	-	√	√	√
*LRIG3*	leucine-rich repeats and immunoglobulin-like domains 3	√	-	√	√	√
*IL1RAPL1*	glutamate decarboxylase-like 1	√	√	√	-	√
*GADL1*	interleukin 1 receptor accessory protein-like 1	-	√	√	√	√
*C5orf22*	chromosome 2 open reading frame, human C5orf22	√	-	√	√	√
*UGT8*	UDP glycosyltransferase 8	√	-	√	√	√
*WISP1*	WNT1-inducible signaling pathway protein 1	√	-	√	√	√
*WNT9A*	Wnt family member 9A	-	√	√	√	√

## Data Availability

The DNA sequencing data of 19 LXFC for this study can be downloaded from the National Genomics Data Center of China (GSA numbers: CRA009027; repository url: https://download.cncb.ac.cn/gsa2/CRA009027/, accessed on 8 June 2022). The RNA sequencing data for this study can be downloaded from the National Genomics Data Center of China (GSA numbers: CRA009037; repository url: https://download.cncb.ac.cn/gsa2/CRA009037/, accessed on 8 June 2022).

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
