# Peer review of "Integrating Whole-Genome Resequencing and RNA Sequencing Data Reveals Selective Sweeps and Differentially Expressed Genes Related to Nervous System Changes in Luxi Gamecocks"

_genes, 2023, doi:10.3390/genes14030584_

Round 1
Reviewer 1 Report
This study integrated the whole-genome sequencing data and RNA-Sequencing data of Luxi gamecock to study the genetics and selection of Luxi gamecock. By performing genome-wide selection sweep scans, they detected a substantial number of genes related to nervous system development that were under selection. By using the RNA-Sequencing data, they conducted differential gene expression analysis, they identified many differential expressed genes (DEGs) between Luxi gamecocks and other gamecocks, where DEGs were related to aggressiveness-related neurotransmitters. They overlapped the genes under selection and DEGs, and found that overlapped genes are associated with nervous system changes in the Luxi gamecock.
The most novelity part of this study is that it has both whole-genome sequencing data and RNA-Sequencing data (right?), and the authors integrated both of those data to understand the molecular mechanism of aggressiveness of gamecock. The study would be helpful for further study about animal and mankind’s aggressiveness. Overall, the analysis is reliable.
My concerns are listed below.
1. The authors did not highlight the value or novelty of their data in the title, abstract, introduction as well as the discussion section. Previous studies actually sequenced the whole genomes of Luxi gamecock. But the RNA-Sequencing data for Luxi gamecock were under-represented in genetic studies (please confirm!). This study integrated both the WGS data and the RNA-Seq data of Luxi gamecock. Therefore I suggest the authors put some efforts in the highlights of their data through the manuscript. For instance, the title can be like "Integrating whole-genome sequencing data and RNA-Sequencing data reveal selective sweeps and differentially expressed genes related to nervous system changes in the Luxi gamecock".
2. The subtitles or substructures in the result selection sucks. Most of the subtitles in the results selection are similar with that in the method section. For example, the subtitle in line 158, Whole genome resequencing data, is just a repeat of the subtitle in the line 84, DNA sample and genome resequencing data. The subtitle in the result section should be a summary of the corresponding section. Actually, in section 3.1, the authors can interpret the overall pattern of the whole-genome sequencing data and the RNA-Sequencing data (which was in section 3.3), and set the subtitle as "Overview of the whole-genome sequencing data and the RNA-Sequencing data". In section 3.2, "selective sweep" can be like "selective sweep reveal genes are associated with ...".
3. The authors are suggested to polish the language or do english editing through the manuscript.
4. Other issues:
Gene names need to be in italics in the abstract section and through the manuscript.
In the title: "whole genome resequencing" -> "whole-genome sequencing";
"RNA-Seq data" -> "RNA-Sequencing data"; "DEGs": do not abbreviate "DEGs", spell it out.
line 124, spell out ROH.
line 143. Add citation.
Author Response
Thanks for your helpful and detailed comments.
Please see the attachment.

Reviewer 2 Report
The current article depicts the set of genes related to nervous system development, Genes associated with aggressiveness-related neurotransmitters in Chinese game cock Luxi gamecock. The author claims that the study provide more insight about molecular mechanism of aggressiveness of gamecock and be helpful for further study about animal and mankind’s aggressiveness.
However, it is not mentioned how the identified genes may be exploited for practical approach in animal and human. Molecular genetics for Human behavioral study has a vast scope. It is not even clear how the information for selective sweep for identified genes may be helpful for studying aggressiveness.
Author Response

(The authors gave the same response as above.)

Reviewer 3 Report
This manuscript presents the results of genome-wide scale analyses to detect selection signatures in genes associated with aggressiveness in gamecocks. I consider that the results can be of interest to scientists and the general audience. However, some considerations could improve the quality of the manuscript.
1. The last paragraph of the introduction (L:75-82) needs to be rewritten. The authors mention that they have used five methods to detect selective sweeps. However, they do not describe or explain why they needed to use five different methods. Furthermore, I believe that the introduction lacks a sentence/paragraph explaining the main objective of this research. Is the effectiveness of each method, or the detection of putative genes/genome regions under selection?
2. The quality of the figures is very poor, especially figures 1 and 3, it is impossible to visualize the labels. Also, I suggest that the gray background in figure 1F should be removed, an empty background may provide better visualization of the results presented in that figure. Similarly, figures 6C and 6D are hard to visualize.
3. There are several spelling mistakes and misused terms in the manuscript. For instance: L157 "Result", I guess using the plural form is a better choice. The word "gamecock" is written as "gamaock" in the conclusions section. I do not think the correct form is "different express gene" used across the manuscript. I think it is more correct to use "differentially expressed genes". Considering these spelling and grammar mistakes I suggest the authors improve the overall quality of the English and check the use of specific terms. There is a substantial difference between "different gene expression analysis, and "Differential gene expression gene analysis (DEG)".
4. Provide a direct url link to the repository where you deposited the raw sequencing data used in this study.
Author Response

(The authors gave the same response as above.)
